# DiffAutoML: Differentiable Joint Optimization for Efficient End-to-End Automated Machine Learning

## Abstract

The automated machine learning (AutoML) pipeline comprises several crucial components such as automated data augmentation (DA), neural architecture search (NAS) and hyper-parameter optimization (HPO). Although many strategies have been developed for automating each component in separation, joint optimization of these components remains challenging due to the largely increased search dimension and different input types required for each component. While conducting these components in sequence is usually adopted as a workaround, it often requires careful coordination by human experts and may lead to sub-optimal results. In parallel to this, the common practice of *searching* for the optimal architecture first and then *retraining* it before deployment in NAS often suffers from architecture performance difference in the search and retraining stages. An end-to-end solution that integrates the two stages and returns a ready-to-use model at the end of the search is desirable. In view of these, we propose a **diff**erentiable joint optimization solution for efficient end-to-end **AutoML** (DiffAutoML). Our method performs co-optimization of the neural architectures, training hyper-parameters and data augmentation policies in an end-to-end fashion without the need of model retraining. Experiments show that DiffAutoML achieves state-of-the-art results on ImageNet compared with end-to-end AutoML algorithms, and achieves superior performance compared with multi-stage AutoML algorithms with higher computational efficiency. To the best of our knowledge, we are the first to jointly optimize automated DA, NAS and HPO in an en-to-end manner without retraining.

## 1 Introduction

While deep learning has achieved remarkable progress in various tasks such as computer vision and natural language processing, it usually requires tremendous human involvement to design and train a satisfactory deep model for one task (He et al., 2016; Sandler et al., 2018). To alleviate such burden on human users, a dozen of AutoML algorithms are proposed in recent years to enable training a model from data automatically without human experiences, including automated data augmentation (DA), neural architecture search (NAS), and hyper-parameter optimization (HPO) (e.g., Chen et al., 2019; Cubuk et al., 2018; Mittal et al., 2020). These AutoML components are usually developed independently. However, implementing these AutoML components for a specific task in separate stages not only suffers from low efficiency but also leads to sub-optimal results (Dai et al., 2020; Dong et al., 2020). How to achieve full-pipeline "from data to model" automation efficiently and effectively is still a challenging problem.

One main difficulty for achieving automated "from data to model" is how to combine different AutoML components (e.g., NAS and HPO) appropriately for a specific task. Optimizing these components in a joint manner is an intuitive solution but usually suffers from the enormous and impractical search space. Dai et al. (2020) and Wang et al. (2020) introduced pre-trained predictors to achieve the joint optimization of NAS and HPO, and the joint optimization of NAS and automated model compression, respectively. For a new coming task, however, it is usually burdensome to pre-train such a predictor. On the other hand, Dong et al. (2020) investigated the joint optimization between NAS and HPO via differentiable architecture and hyper-parameter search spaces. Automated DA is seldom considered in the joint optimization of AutoML components. Nevertheless, our experimental

results showed that different data augmentation protocols may result in different optimal architectures (see Sec. 4.3 for more details). Based on these considerations, it is worthy to investigating the joint optimization among automated DA, NAS and HPO.

Another main challenge to achieve automated "from data to model" is the end-to-end searching and training of models without parameter retraining. Even considering only one AutoML component, many NAS algorithms require two stages including searching and retraining (e.g., Liu et al., 2018; Xie et al., 2019). Automated DA methods such as Lim et al. (2019) also needed to retrain the model parameters when the DA policies were searched. In these cases, whether the searched architectures or DA policies would perform well after retraining is questionable, due to the inevitable difference of training setup between the searching and retraining stages. Recently, Hu et al. (2020) developed a differentiable NAS methods to provide direct NAS without parameter retraining. Other AutoML components, including HPO and automated DA, are seldom considered in the task-specific end-to-end AutoML algorithms.

Considering the above challenges, we propose DiffAutoML, a differentiable joint optimization solution for efficient end-to-end AutoML. In DiffAutoML, end-to-end NAS optimization is realized in a differentiable one-level manner with the help of stochastic architecture search and the approximation for the gradient of architecture parameters. Meanwhile, the DA and HPO are regarded as dynamic schedulers, which adapt themselves to the update of network parameters and network architecture. Specifically, Differentiable relaxation is used in DA optimization in an one-level way while the hyper-gradient is used to in HPO in a two-level way. With this differentiable method, DiffAutoML can effectively deal with the huge search space and the low optimization efficiency caused by this joint optimization problem. To summarize, our main contributions are as follows:

- We propose a well-defined AutoML problem, i.e., task-specific end-to-end AutoML framework, which aims to achieve automated "from data to model" without human involvement.
- We first jointly optimize three different AutoML components, including automated DA, NAS and HPO, in a differentiable search space with high efficiency.
- Experiment results show that, compared with optimizing modules in sequence, one-stage DiffAutoML can effectively realize the co-optimization of different modules.
- Extensive experiments also reveal mutual influence among different AutoML components, i.e., the change of settings for one module may greatly influence the optimal results of another module, justifying the necessity of end-to-end joint optimization.

## 2 RELATED WORKS

In this section, we briefly introduce some related AutoML algorithms, including automated DA, NAS and HPO. A more detailed introduction is provided in Appendix D.

**Automated Data Augmentation** Data augmentation is commonly used to increase the data diversity and thus to improve the generalization performance of deep learning models (DeVries & Taylor, 2017; Zhang et al., 2017; Yun et al., 2019). Several prior works have been proposed to automate the search of data augmentation policies, using methods ranging from reinforcement learning (Cubuk et al., 2018; Zhang et al., 2020), evolutionary algorithm (Ho et al., 2019), Bayesian optimization (Lim et al., 2019), gradient-based approaches (Lin et al., 2019) to simple grid search (Cubuk et al., 2020). Cubuk et al. (2020) point out that the optimal data augmentation policy depends on the model size. This indicates that fixing an augmentation policy when searching for neural architectures may lead to sub-optimal solutions, thus motivating joint optimization.

**Neural Architecture Search** Recent advances in NAS have demonstrated state-of-the-art performance outperforming human experts' design on a variety of tasks (Zoph & Le, 2017; Cai et al., 2018; Liu et al., 2018; Kandasamy et al., 2018; Pham et al., 2018; Real et al., 2018; Zoph et al., 2018; Ru et al., 2020b). Especially the development of gradient-based one-shot methods (Liu et al., 2018; Chen et al., 2019; Guo et al., 2019; Xie et al., 2019; Hu et al., 2020) have significantly reduced the computational costs of NAS. To further improve the search efficiency, most NAS methods search architectures in a low-fidelity set-up (e.g., fewer training epochs, smaller architectures) and retrain the optimal architecture using the full set-up before deployment. This separation of *search* and *evaluation*, however, usually suffer from sub-optimal results (Hu et al., 2020). End-to-end NAS

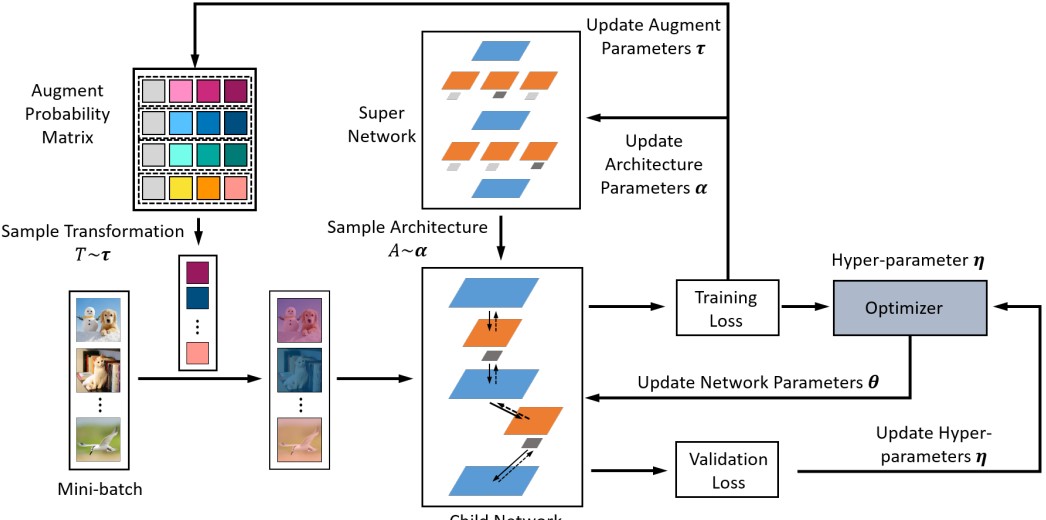

Figure 1: An overview of DiffAutoML. We first sample the DA operations for each sample based on the data transformation parameters $\tau$. Then, a child network is sampled based on the architecture parameters $\alpha$, which will be used to process transformed mini-batch. Training loss is calculated to update the neural network parameter $\theta$ by an optimizer with the hyper-parameters $\eta$. Parameters $\tau$ and $\alpha$ are updated based on the training loss, while $\eta$ is updated with the validation loss to achieve differentiable end-to-end joint optimization of automated DA, NAS and HPO.

strategies (Xie et al., 2019; Hu et al., 2020) are thereby developed to return read-to-deploy networks at the end of the search. Our work also proposes an end-to-end solution.

**Hyper-parameter Optimization** Tuning hyper-parameters for deep learning models has also attracted lots of research attention. Current HPO methods address various set-ups involving multi-fidelity evaluations, parallel computation, transfer-learning (Mittal et al., 2020) or a mixed variable types (Ru et al., 2020a). Although many HPO strategies have been adapted to form NAS strategies, methods that jointly optimize both architectures and hyper-parameters are rarely seen except the ones discussed below.

**Joint Optimization of AutoML Components.** Conventional neural architecture search methods perform search under a fixed set of training hyper-parameters and apply or search for a separate set of hyper-parameters when retraining the best architecture found. Such search protocol may lead to sub-optimal results (Zela et al., 2018; Dong et al., 2020) as it neglects the influence of training hyper-parameters on architecture performance and ignores superior architectures under alternative hyper-parameter values (Dai et al., 2020). In view of this, several works have been proposed to jointly optimise architecture structure and training hyper-parameters (Dai et al., 2020; Wang et al., 2020; Dong et al., 2020). Zela et al. (2018) introduces the use of multi-fidelity Bayesian optimization to search over both the architecture structure and training hyper-parameters. Dai et al. (2020) trains an accuracy predictor to estimate the network performance based on both the architecture and training hyper-parameters, and then uses evolutionary algorithm to perform the search. Both these methods are sample-based and requires a relatively large number of architecture and hyper-parameter evaluations to finetune predictor or obtain good recommendations. A more related work is AutoHAS (Dong et al., 2020), which introduces a differentiable approach in conjunction with weight sharing for the co-optimization task. Dong et al. (2020) demonstrates empirically that such differentiable one-shot approach achieve superior efficiency over sampled-based methods. Our proposed method differs from AutoHAS in mainly two aspects: first, AutoHAS updates the entire supernetwork at each optimization step while our method trains only a subnetwork. Thus, our method is much more memory efficient than AutoHAS. Second, we further extend the co-optimization scope from NAS and training hyper-parameters to also include data augmentation hyper-parameters. To the best of our knowledge, we are the first to jointly optimize all three aspects and we demonstrate the advantage of such practice in Sec. 4.

## 3 METHODOLOGY

Consider a dataset $\mathcal{D} = \{(x_i, y_i)\}_{i=1}^N$, where $N$ is the size of this dataset, and $y_i$ is the label of the input sample $x_i$. We aim to train a neural network $f(\cdot)$, which can achieve the best accuracy on the test dataset $\mathcal{D}^{test}$. Multiple AutoML components are considered, including automated DA, NAS, and HPO. Let $\boldsymbol{\tau}$, $\boldsymbol{\alpha}$, $\boldsymbol{\eta}$, and $\boldsymbol{\theta}$ represent the data augmentation parameters, the architecture parameters, the hyper-parameters, and the objective neural network parameters, respectively. This problem can be formulated as

$$\underset{\boldsymbol{\tau}, \boldsymbol{\alpha}, \boldsymbol{\eta}, \boldsymbol{\theta}}{\arg \min} \mathcal{L}(\boldsymbol{\tau}, \boldsymbol{\alpha}, \boldsymbol{\eta}, \boldsymbol{\theta}; \mathcal{D})$$
$$s.t. \quad c_i(\boldsymbol{\alpha}) \le C_i, i = 1, ..., \gamma, \tag{1}$$

where $\mathcal{L}(\cdot)$ represents the loss function; $\mathcal{D}$ denotes the input data; $c_i(\cdot)$ and $\gamma$ refer to the formula and count of resource constraints $C_i$, such as storage cost and computational cost. We adopt different loss functions to optimize the corresponding parameters, which will be introduced in the following subsections. Considering the huge search space, it is challenging to achieve the co-optimization of $\boldsymbol{\tau}$, $\boldsymbol{\alpha}$, $\boldsymbol{\eta}$, and $\boldsymbol{\theta}$ within one-stage without parameter retraining. In this work, we propose to use the differentiable method to provide a computationally efficient solution. See Fig. 1 as an illustration.

### 3.1 OPTIMIZATION OF DATA AUGMENTATION PARAMETERS

For every mini-batch of training data $\mathcal{B}^{tr} = \{(x_k, y_k)\}_{k=1}^{n^{tr}}$ with batch size $n^{tr}$, we conduct data augmentation to increase the diversity of the training data. We consider $K$ data augmentation operations, and each training sample is augmented by a transformation consisting of two successive operations (Cubuk et al., 2018; Lim et al., 2019). Each operation is associated with a magnitude that is uniformly sampled from $[0, 10]$. The data augmentation parameter $\boldsymbol{\tau}$ represents a probability distribution over the augmentation transformations. For $t$-th iteration, we sample $n^{tr}$ transformations according to $\boldsymbol{\tau}^t$ with Gumbel-Softmax reparameterization (Maddison et al., 2016) and to generate the corresponding augmented samples in the batch. Given a sampled architecture (see details in Sec. 3.2), the loss function for each augmented sample is denoted by $\mathcal{L}^{tr}(f(\boldsymbol{\alpha}^t, \boldsymbol{\theta}^t; \mathcal{T}_k(x_k)))$, where $\mathcal{T}_k$ represents the selected transformation. In order to relax $\boldsymbol{\tau}$ to be differentiable, we regard $p_k(\boldsymbol{\tau}^t)$, the probability of sampling the transformation $\mathcal{T}_k$, as an importance weight for the loss function $\mathcal{L}^{tr}(f(\boldsymbol{\alpha}^t, \boldsymbol{\theta}^t; \mathcal{T}_k(x_k)))$. The objective of data augmentation is to minimize the following loss function:

$$\mathcal{L}^{DA}(\boldsymbol{\tau}^t) = -\sum_{k=1}^{n^{tr}} p_k(\boldsymbol{\tau}^t) \mathcal{L}^{tr}(f(\boldsymbol{\alpha}^t, \boldsymbol{\theta}^t; \mathcal{T}_k(x_k))), \tag{2}$$

With this loss function, DiffAutoML intends to increase the sampling probability of those transformations that can generate samples with high training losses. By sampling such transformations, DiffAutoML can pay more attention to relatively "hard" samples and increase model robustness against difficult samples. Instead of training a controller to generate adversarial augmentation policies via reinforcement learning (Zhang et al., 2020), we search for the probability distribution of augmentation transformations directly via gradient-based optimization. In this way, the optimization of data augmentation is very efficient and hardly increases the computing cost.

### 3.2 OPTIMIZATION OF ARCHITECTURE PARAMETERS

With the augmented data in Sec. 3.1, we achieve optimization of the architecture parameter $\boldsymbol{\alpha}$ through end-to-end NAS, motivated by SNAS (Xie et al., 2019) and DSNAS (Hu et al., 2020). Following Liu et al. (2018), we denote the neural architecture search space as a single directed acyclic graph (DAG). Node $m_i$ refers to the feature map in DAG and edge $(m_i, m_j)$ refers to the information flow between $m_i$ and $m_j$. In addition, $\mathbf{O}_{i,j}$ and $\boldsymbol{\alpha}_{i,j}$ respectively refer to possible operations and the architecture parameters between $m_i$ and $m_j$, and $\boldsymbol{\alpha}$ consists of all possible $\boldsymbol{\alpha}_{i,j}$ in the DAG. During the forward process, an ont-hot random variable $\boldsymbol{Z}_{i,j}$ is sampled from a distribution defined by the architecture parameters $\boldsymbol{\alpha}_{i,j}$. Then, the sampled operation $\tilde{\boldsymbol{O}}_{i,j}$ can be calculated by multiply $\boldsymbol{Z}_{i,j}$ to $\boldsymbol{O}_{i,j}$. Thus, the intermediate node $m_j$ can be expressed as:

$$m_j = \sum_{i<j} \tilde{\boldsymbol{O}}_{i,j} = \sum_{i<j} \boldsymbol{Z}_{i,j}^T \boldsymbol{O}_{i,j}. \tag{3}$$

Instead of using validation loss, the architecture parameters is optimized using the training loss. In that way, architecture parameter $\boldsymbol{\alpha}$ and parameter $\boldsymbol{\theta}$ can be trained with the same loss function. SNAS relaxes $\boldsymbol{Z}_{i,j}$ to a continuous random variable $\tilde{\boldsymbol{Z}}_{i,j}$ with the Gumble-Softmax. Thus gradient of $\boldsymbol{\alpha}$ under training loss can be formulated as:

$$\frac{\partial \mathcal{L}^{tr}}{\partial \alpha_{i,j}^k} = \frac{\partial \mathcal{L}^{tr}}{\partial m_j} \boldsymbol{O}_{i,j}^T(m_i)(\boldsymbol{\delta}(k'-k) - \tilde{\boldsymbol{Z}}_{i,j}) Z_{i,j}^k \frac{1}{\lambda \alpha_{i,j}^k}, \tag{4}$$

where $\alpha_{i,j}^k$ is the $k$-th element in $\boldsymbol{\alpha}_{i,j}$; $Z_{i,j}^k$ is the $k$-th element in $\boldsymbol{Z}_{i,j}$; $\lambda$ is the temperature in the Gumbel-Softmax. As $\lambda$ is the denominator in the gradient, it cannot be zero, which prevents SNAS from realizing the architecture search and the network parameter tuning in an end-to-end manner. In order to realize the one-stage optimization, we adopt the following Eqn. 5 to approximate its gradient:

$$\lim_{\lambda \to 0} \mathbb{E}_{\tilde{\boldsymbol{Z}} \sim p(\tilde{\boldsymbol{Z}})} \left[ \frac{\partial \mathcal{L}^{tr}}{\partial \alpha_{i,j}^k} \right] = \mathbb{E}_{\boldsymbol{Z} \sim p(\boldsymbol{Z})} \left[ \nabla_{\alpha_{i,j}^k} \log p(\boldsymbol{Z}_{i,j}) \frac{\partial \mathcal{L}^{tr}}{\partial Z_{i,j}^s} \right], \tag{5}$$

where $Z_{i,j}^s$ is the $s$-th element in the one-hot vector $\boldsymbol{Z}_{i,j}$ where $s$-th element is equal to one.

### 3.3 OPTIMIZATION OF HYPER-PARAMETERS

As shown in Fig. 1, given the batch of augmented training data $\{(\mathcal{T}_k(x_k), y_k)\}_{k=1}^{n^{tr}}$ and the sampled child network, we need to optimize the differentiable hyper-parameters $\boldsymbol{\eta}$, such as learning rate and L2 regularization. At the training stage, we alternatively update $\boldsymbol{\theta}$ and $\boldsymbol{\eta}$. In $t$-th iteration, we can update $\boldsymbol{\theta}^t$ based on the gradient of the training loss $\mathcal{L}^{tr}(f(\boldsymbol{\alpha}^t, \boldsymbol{\theta}^t; \mathcal{B}^{tr})) = \frac{1}{n^{tr}} \sum_{k=1}^{n^{tr}} \mathcal{L}^{tr}(f(\boldsymbol{\alpha}^t, \boldsymbol{\theta}^t; \mathcal{T}_k(x_k)))$, which can be written as:

$$\boldsymbol{\theta}^{t+1} = OP(\boldsymbol{\theta}^t, \boldsymbol{\eta}^t, \nabla_{\boldsymbol{\theta}} \mathcal{L}^{tr}(f(\boldsymbol{\alpha}^t, \boldsymbol{\theta}^t; \mathcal{B}^{tr}))), \tag{6}$$

where $OP(\cdot)$ is the optimizer. To update the hyper-parameters $\boldsymbol{\eta}$, we regard $\boldsymbol{\theta}^{t+1}$ as a function of $\boldsymbol{\eta}$ and compute the validation loss $\mathcal{L}^{val}(f(\boldsymbol{\alpha}^t, \boldsymbol{\theta}^{t+1}(\boldsymbol{\eta}^t); \mathcal{B}^{val}))$ with network parameters $\boldsymbol{\theta}^{t+1}(\boldsymbol{\eta}^t)$ on a mini-batch of validation data $\mathcal{B}^{val}$. Then, $\boldsymbol{\eta}^t$ is updated with $\nabla_{\boldsymbol{\eta}} \mathcal{L}^{val}(f(\boldsymbol{\alpha}^t, \boldsymbol{\theta}^{t+1}(\boldsymbol{\eta}^t); \mathcal{B}^{val}))$ by gradient descent:

$$\boldsymbol{\eta}^{t+1} = \boldsymbol{\eta}^t - \beta \nabla_{\boldsymbol{\eta}} \mathcal{L}^{val}(f(\boldsymbol{\alpha}^t, \boldsymbol{\theta}^{t+1}(\boldsymbol{\eta}^t); \mathcal{B}^{val})), \tag{7}$$

where $\beta$ is a learning rate. Even $\boldsymbol{\theta}^t$ can also be deployed to $\boldsymbol{\theta}^{t-1}$ whose calculation also involves $\boldsymbol{\eta}$, we take an approximation method in this equation and regard $\boldsymbol{\theta}^t$ here as a variable independent of $\boldsymbol{\eta}$.

### 3.4 DIFFAUTOML

Based on above analysis of each AutoML module, DiffAutoML realizes end-to-end co-optimization of automated data augmentation parameters $\boldsymbol{\tau}$, architecture parameters $\boldsymbol{\alpha}$, and hyper-parameters $\boldsymbol{\eta}$. The DiffAutoML algorithm is summarized in Algorithm 1. One-level optimization is applied to $\boldsymbol{\alpha}$ and $\boldsymbol{\tau}$ as in Line 5 and Line 6, while bi-level optimization is applied to $\boldsymbol{\eta}$ as in Line 8.

## 4 EXPERIMENTS

Experiments are conducted to verify the performance of the proposed DiffAutoML. In Sec. 4.2, we compare the performance of DiffAutoML with end-to-end NAS and multi-stage NAS in terms of final accuracy and the computational efficiency to show the advantages of end-to-end joint optimization. In Sec. 4.3, we conduct ablation studies for DiffAutoML. First, given neural network architectures, we verify the effectiveness of the automated DA and HPO modules provided by DiffAutoML. Second, we investigate the mutual influence among different AutoML components.

### 4.1 EXPERIMENT SETTING

**Dataset** All our experiments are conducted on the ImageNet dataset (Russakovsky et al., 2015) for the classification task. This dataset consists of around $1.28 \times 10^6$ training images and $5 \times 10^4$ validation images.

---

**Algorithm 1** DiffAutoML

---

**Initialization:** Architecture Parameters $\alpha$, Data Transformation Parameters $\tau$, Hyper-parameters $\eta$ and Network Parameters $\theta$
**Input:** Training Set $\mathcal{D}^{tr}$, Validation Set $\mathcal{D}^{val}$, Optimizer, Parameters $\tau$, $\alpha$, $\eta$, $\theta$, and the iteration number $T$
**Return:** $\tau, \alpha, \eta, \theta$

1: **while** $t < T$ **do**
2:      Separately sample a mini-batch $\mathcal{B}^{tr}$ and a mini-batch $\mathcal{B}^{val}$ from $\mathcal{D}^{tr}$ and $\mathcal{D}^{val}$;
3:      For each sample in $\mathcal{B}^{tr}$, sample a transformation according to $\tau^t$;
4:      Based on $\alpha^t$, sample a child network from super network;
5:      Compute the weighted loss function as Eqn. (2) and update $\tau^{t+1}$ accordingly;
6:      Use normal loss function to update $\alpha^{t+1}$ with Eqn. (5);
7:      Calculate $\theta^{t+1}$ with Eqn. (6);
8:      Compute the loss function with $\theta^{t+1}$ on $\mathcal{D}^{val}$ and update $\eta^{t+1}$ with Eqn. (7);

---

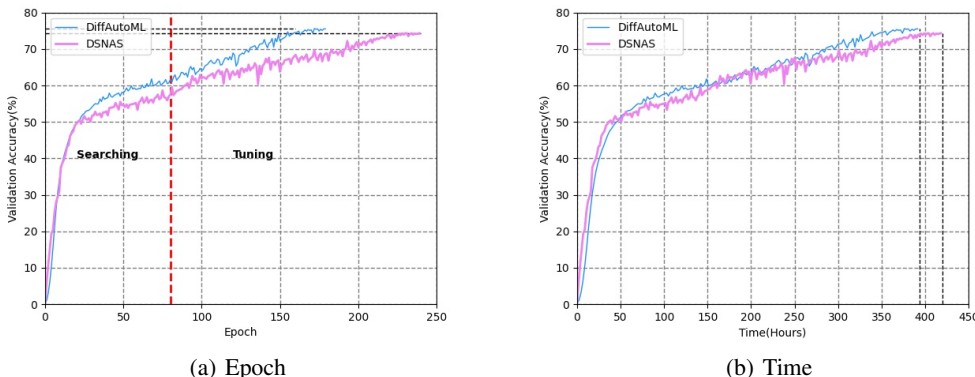

(a) Epoch                      (b) Time

Figure 2: Comparison of validation accuracy of DSNAS and DiffAutoML over (a) the training epochs and (b) the run time. NAS optimization in DiffAutoML can be divided into searching stage where $\alpha$ is optimized with others, and tuning stage where $\alpha$ is fixed while other modules are still updated.

**Search Space   Automated DA.** Following Ho et al. (2019), we consider 14 different operations for data augmentation, including AutoContrast, Equalize, Rotate, Posterize, Solarize, Color, Contrast, Brightness, Sharpness, Shear X/Y, Translate X/Y, and Identity. The magnitude of each operation is randomly sampled from the uniform distribution. **NAS.** Following Hu et al. (2020), we consider four candidates for each choice block in the parent network, i.e., choice_1, choice_2, choice_3, and choice_4. These candidates differ in kernel size and the number of depthwise convolutions (see Appendix B for the detailed network architectures). The resulting search space consists of $4^{20}$ single path models. **HPO.** For illustration purpose, we consider the L2 regularization (i.e., weight decay) in the experiments.

### 4.2   Joint Optimization of AutoML Components

We first compare our DiffAutoML against several types of baseline methods: memory-efficient manual designed networks such as MobileNet V2 (1.0x) (Sandler et al., 2018) and ShuffleNet V2 (1.5) (Ma et al., 2018), conventional two-stage NAS methods like Proxyless-R (mobile) (Cai et al., 2019) and Single Path One-Shot (Guo et al., 2019), which optimize the architectures during a *search* stage and then *retrain* the best architecture before deployment, and finally the recently proposed end-to-end NAS approach, DSNAS (Hu et al., 2020). Detailed experimental settings are provided in Appendix A.

The results of this set of experiments are summarized in Table 1. First, it is evident that all NAS methods including our DiffAutoML clearly outperform the manually designed networks both in terms of accuracy performance as well as FLOPS required; this verifies the effectiveness of performing architecture search. Second, if we take into account the total time taken for obtaining a

ready-to-deploy network [1], the end-to-end NAS approaches (DSNAS and DiffAutoML) consumes less time than two-stage NAS approaches (Proxyless-R and Single Path One-Shot) to achieve better or comparable performance. Specifically, DiffAutoML exceeds Proxyless-R by $0.9\%$ in terms of accuracy without retraining but takes $42\%$ less time. Note for end-to-end NAS methods, the accuracy performance of the network found with and without training are almost identical. Finally, the performance gain of our DiffAutoML over DSNAS shows the advantage of doing joint optimization of automated DA, NAS and HPO in comparison with doing NAS alone. In addition, we compare the validation accuracy of DiffAutoML and DSNAS over epoch numbers (Fig. 2(a)) and training time (Fig. 2(b)). As can be seen, DiffAutoML achieves higher validation accuracy with much fewer epochs. Although optimizing more AutoML components requires slightly longer time for each epoch, DiffAutoML results in less total searching time due to the fast convergence compared to DSNAS (see Fig. 2(b)). This indicates that appropriate automated DA and HPO could help to accelerate and stabilize the NAS process, and also shows the efficiency of our differentiable joint optimization framework even with a much larger search dimension.

Table 1: Top-1 Accuracy ($\%$) and Top-5 Accuracy ($\%$) of architecture search algorithms in the ImageNet dataset. Here, * indicates our application, ‡ represents the accuracy of the search set, and § refers to a conversion from V100 GPU hours with a ratio of $1 : 1.5$.

| Model | FLOPS | Search | | Retrain | | Time (GPU hour) | |
|---|---|---|---|---|---|---|---|
| | | Top-1 Acc(%) | Top-5 Acc(%) | Top-1 Acc(%) | Top-5 Acc(%) | Search | Retrain |
| MobileNet V2 (1.0x) | 564M | Manual | | 72.0 | 91.0 | Manual | |
| ShuffleNet V2 (1.5x) | 564M | Manual | | 72.6 | 90.6 | Manual | |
| Proxyless-R (mobile) | 320M | 62.5* | 84.8* | 74.6 | 92.2 | 300§ | ≥384 |
| Single Path One-Shot | 319M | 68.7‡ | - | 74.3 | - | 250 | 384 |
| DSNAS | 324M | 74.4 ± 0.2 | 91.5 | 74.3 ± 0.3 | 91.9 | 420 | |
| DiffAutoML | 318M | **75.5** | **92.7** | **75.5** | **92.5** | 394 | |

## 4.3 ABLATION STUDIES AND DISCUSSION

### 4.3.1 DIFFAUTOML GIVEN NETWORK ARCHITECTURES

We conduct this set of experiments to demonstrate the effectiveness of DA and HPO modules in DiffAutoML, as well as the advantages of conducting optimization of AutoML modules jointly over sequentially.

We first apply the joint optimization of automated DA and HPO (see Sec. 3.1 and 3.3) in the retraining phases of the baseline NAS algorithms including DARTS and DSNAS. This is to simulate the common practice of optimizing different components of AutoML separately in sequence. The results in the top two rows of Table 2 show that performance gain can be obtained by doing such additional optimization on HP and DA policies, justifying the need of applying multiple AutoML components for a single task. It also shows that the DA and HPO optimization modules in our method remain performant even when being used in separation from our NAS approach. Finally, compared with the performance reported in Table 1, the superior performance of doing co-optimization of all the components (i.e., automated DA, NAS and HPO) over doing optimization in sequence shows the clear advantage of joint optimization.

We also apply joint optimization of automated DA and HPO provided by DiffAutoML to manual design networks such as ResNet-50. There is one thing worth mentioning that in this experiment, the HPO optimizes both learning rate and L2 regularization. The results are compared with recent automated DA algorithms including Fast Autoaugment (FAA) (Lim et al., 2019), Population-Based Augment (PBA) (Ho et al., 2019) and Adversarial Autoaugment (AdvAA), as summarized in the last seven rows of Table 2. As can be seen, without delicate hyper-parameter tuning with burdensome human involvement, our DiffAutoML shows very competitive results, outperforming the current baselines in most cases on a variety of image classification tasks.

---

[1]The sum of search and retrain time for two-stage NSA methods and the search time along for the end-to-end NAS method.

Table 2: Top-1 / Top-5 Accuracy (%) of different DA algorithms and the NAS algorithms on ImageNet. Top-1 Accuracy (%) of different DA algorithms on CIFAR-10 and CIFAR-100.

| Dataset | Model | Baseline | FAA | PBA | AdvAA | DiffAutoML |
|---|---|---|---|---|---|---|
| ImageNet | DARTS (DA+HPO) | 73.1/91.0 | – | – | – | **73.8/91.5** |
| | DSNAS (DA+HPO) | 74.3/91.9 | – | – | – | **75.0/92.5** |
| ImageNet | ResNet-50 | 76.3/93.1 | 77.6/93.7 | – | **79.4/94.5** | 79.0/94.3 |
| | ResNet-200 | 78.5/94.2 | 80.6/95.3 | – | 81.3/95.3 | **81.5/95.5** |
| CIFAR-10 | Wide-ResNet-28-10 | 96.1 | 97.3 | 97.4 | **98.1** | $98.0 \pm 0.1$ |
| | Wide-ResNet-40-2 | 94.7 | 96.4 | – | – | $\mathbf{96.8 \pm 0.0}$ |
| | PyramidNet+ShakeDrop | 97.3 | 98.3 | 98.5 | 98.6 | $\mathbf{98.6 \pm 0.1}$ |
| CIFAR-100 | Wide-ResNet-28-10 | 81.2 | 82.9 | 83.3 | **84.5** | $84.0 \pm 0.2$ |
| | Wide-ResNet-40-2 | 74.0 | 79.3 | – | – | $\mathbf{80.6 \pm 0.0}$ |

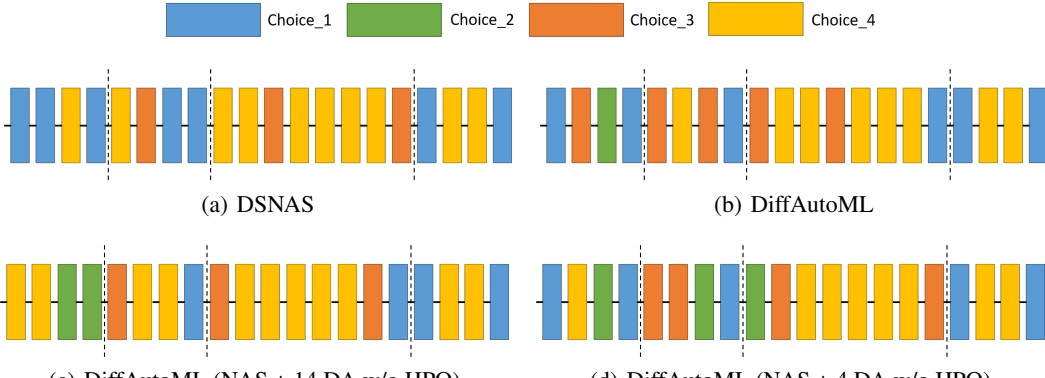

(a) DSNAS  (b) DiffAutoML

(c) DiffAutoML (NAS + 14 DA w/o HPO)  (d) DiffAutoML (NAS + 4 DA w/o HPO)

Figure 3: Searched architectures with different AutoML components. The network architecture is divided into four components shown in the above figure with the help of the dotted line. The first block of each component has the stride number of 2 and the rest blocks have the stride number of 1. Detailed architectures of different choices are provided in Appendix B.

### 4.3.2 MUTUAL INFLUENCE AMONG AUTOML COMPONENTS

In this experiment, we investigate the mutual inference among different AutoML components by comparing the searched architectures and the final accuracy. We first pay attention to the resulting architectures of DSNAS and DiffAutoML in Sec. 4.2. For comparison purpose, we also consider different DA strategies in NAS without HPO when using DiffAutoML. Specifically, we separately conduct DiffAutoML with all DA operations introduced in Sec. 4.1 and four DA operations only. The four DA operations are Shear X/Y and Translate X/Y.

The resulting architectures are provided in Fig. 3, and the corresponding final accuracy is provided in Appendix C. As can be seen, the resulting optimal architectures are quite different in terms of choices of basic building blocks. This indicates mutual influences among different AutoML components, thus justifying the need for co-optimization of AutoML components instead of optimizing them separately.

## 5 CONCLUSION

Previous AutoML algorithms usually focus on one or two components only, which ignore the coupling relationships among different AutoML components. DiffAutoML takes the co-optimization of the data augmentation, the architecture search, and the hyper-parameters into consideration, adapts the differentiable method to well treat this large search space, and realizes the end-to-end co-optimization. With less computation complexity, DiffAutoML achieves better results than alternative one-stage NAS and comparable results with two-stage NAS.

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

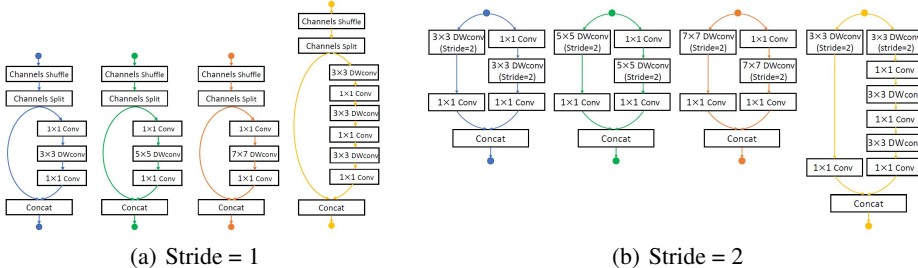

(a) Stride = 1            (b) Stride = 2

Figure 4: Choice blocks in search space. From left to right are Choice_1, Choice_2, Choice_3, and Choice_4.

## A    DETAILED EXPERIMENTAL SETTINGS

This experiments are run on 8*NVIDIA V100 under PyTorch-1.3.0 and python3.6. Hyper-parameter $\eta$ is optimized by the Adam optimizer with the learning rate $1 \times 10^{-3}$. The data transformation parameter $\tau$ and the architecture parameter $\eta$ are optimized by the SGD optimizer separately with the learning rate $1 \times 10^{-5}$ and $1 \times 10^{-6}$. Their L2 weight decay is set to be $5 \times 10^{-4}$ and their momentum is set to be $0.9$. SGD optimizer is used to optimize $\theta$. Cosine learning scheduler is used to gradually change the learning rate with the initial learning rate $0.5$ and the moment of SGD optimizer is set to $0.9$.

As to the optimization of $\tau$ and $\eta$ with given architectures, we use the same optimizer and hyper-parameter as in Sec. 4.2. For the retraining of the architecture found by DSNAS, the SGD optimizer is used with the basic learning rate $0.5$ and the L2 weight decay $4 \times 10^{-5}$. The learning rate is also gradually modified by the cosine learning scheduler. The retraining of DARTS shares almost the same setting as that DSNAS, while its initial learning rate is set to $0.1$ with the weight decay $3 \times 10^{-5}$.

## B    DETAILS ABOUT THE ARCHITECTURES

Following Hu et al. (2020), we consider four candidates for each choice block in the parent network, i.e., Choice_1, Choice_2, Choice_3, and Choice_4. These candidates differ in kernel size and the number of depthwise convolutions. Detailed network architectures are provided in Fig. 4.

## C    MUTUAL INFLUENCE AMONG AUTOML COMPONENTS

In this experiment, we investigate the mutual inference among different AutoML components by comparing the searched architectures and the final accuracy. For comparison purpose, we consider different DA strategies in NAS without HPO when using DiffAutoML. Specifically, we conduct DiffAutoML with all DA operations introduced in Sec. 4.1 and four DA operations only, separately. The four DA operations are Shear X/Y and Translate X/Y. The resulting accuracy in ImageNet is reported in Table 3.

Table 3: The Top-1 Accuracy (%) and Top-5 Accuracy (%) in the ImageNet dataset achieved by DiffAutoML trained under different DA settings.

| Method | Top-1 Acc(%) Co-optimization | Top-5 Acc(%) Co-optimization | FLOPS |
|---|---|---|---|
| DiffAutoML (NAS + 4 DA) | 74.8 | 92.2 | 323M |
| DiffAutoML (NAS + 14 DA) | 75.1 | 92.7 | 318M |

## D    RELATED WORK IN DETAILS

### D.1    NEURAL ARCHITECTURE SEARCH

NAS is an important subfield of AutoML that automates the design of neural networks for various tasks. It has attracted growing attention over the recent years and discovered architectures with better performance over those designed by human experts (Zoph & Le, 2017; Cai et al., 2018; Liu et al., 2018; Kandasamy et al., 2018; Pham et al., 2018; Real et al., 2018; Zoph et al., 2018; Ru et al., 2020b). The rich collection of NAS literature can be divided into two categories: the query-based methods and the gradient-based ones. The former includes powerful optimization strategies such as reinforcement learning (Zoph & Le, 2017; Pham et al., 2018), Bayesian optimization (Kandasamy et al., 2018; Ru et al., 2020b) and evolutionary algorithms (Elsken et al., 2019; Lu et al., 2019). The latter enables the use of gradients in updating both architecture parameters and network weights, and significantly reduces the computation costs of NAS via weight sharing. Since the seminal work DARTS (Liu et al., 2018), many follow-up works (Chen et al., 2019; Xie et al., 2019; Hu et al., 2020) have been proposed to improve the performance and efficiency of gradient-based approaches. To achieve cost efficiency, most NAS methods search architectures in a low-fidelity set-up (e.g.fewer training epochs, smaller architectures) and retrain the optimal architecture using the full set-up before deployment. This separation of *search* and *evaluation* is sub-optimal (I cannot find the exact drawbacks of this except something like poor performance ranking correlation between the architectures at the end of searched and after retraining) (Hu et al., 2020) and motivates the development of end-to-end NAS strategies (Xie et al., 2019; Hu et al., 2020) which return read-to-deploy networks at the end of the search. Our work also proposes an end-to-end solution.

### D.2    DATA AUGMENTATION

Data augmentation is one of the commonly used techniques in deep learning to increase the diversity of data and thus to improve the generalization ability of deep learning models. Elaborately designed augmentation methods have effectively improved the performance of deep models (DeVries & Taylor, 2017; Zhang et al., 2017; Yun et al., 2019). However, the effectiveness of these manually designed ones varies among datasets. As automated machine learning has achieved impressive success in quite a lot scenarios, learning data augmentation policies directly from a target dataset has become a trend. AutoAugment (Cubuk et al., 2018) and Adversarial AutoAugment (Zhang et al., 2020) adopt reinforcement learning to train a controller to generate policies, while OHL-Auto-Aug (Lin et al., 2019) formulates augmentation policy as a probability distribution and adopts REINFORCE (Williams, 1992) to optimize the distribution parameters along with network training. PBA (Ho et al., 2019) and FAA (Lim et al., 2019) use population-based training method and Bayesian optimization respectively to reduce the computing cost of learning policies. Cubuk et al. (2020) argue that the search space of policies used by these works can be reduced greatly and simple grid search can achieve competitive performance. They also point out that the optimal data augmentation policy depends on the model size, which indicates that fixing an augmentation policy when searching for neural architectures may lead to sub-optimal solutions.

### D.3    HYPER-PARAMETER OPTIMIZATION

The performance of neural networks heavily depends on the appropriate choices of hyper-parameters. Various black-box optimization approaches have been developed to address hyper-parameter tuning tasks involving multiple tasks (Mittal et al., 2020) or a mixed variable types (Ru et al., 2020a). Meanwhile, techniques like multi-fidelity evaluations, parallel computation and transfer learning are also employed to further enhance the query efficiency of the hyper-parameter optimization. Although many hyper-parameter optimization strategies have been adapted to form query-based NAS strategies, methods that jointly optimize both architectures and hyper-parameters are rarely seen except the ones discussed below.

### D.4    CO-OPTIMIZATION

Conventional neural architecture search methods perform search under a fixed set of training hyper-parameters and apply or search for a separate set of hyper-parameters when retraining the best architecture found. Such search protocol leads to sub-optimal results (Zela et al., 2018; Dong et al.,

2020) as it overlooks the influence of training hyper-parameters on architecture performance and ignores superior architectures under alternative hyper-parameter values (Dai et al., 2020). In view of this, several works have been proposed to jointly optimize architecture structure and training hyper-parameters (Dai et al., 2020; Wang et al., 2020; Dong et al., 2020). Zela et al. (2018) introduces the use of multi-fidelity Bayesian optimization to search over both the architecture structure and training hyper-parameters. Dai et al. (2020) trains an accuracy predictor to estimate the network performance given its architecture and training hyper-parameters and then uses the predictor with evolutionary algorithm to perform the search. Both these methods are sample-based and requires a relatively large number of architecture-hyperparameter evaluations to finetune predictor or obtain good recommendations. A more related work is AutoHAS (Dong et al., 2020), which introduces a differentiable approach in conjunction with weight sharing for the co-optimization task. Dong et al. (2020) demonstrates empirically that such differentiable one-shot approach achieve superior efficiency over sampled-based methods. Our proposed method differs from AutoHAS in mainly two aspects: first, AutoHAS updates the entire supernetwork at each optimization step while our method trains only a subnetwork. Thus, our method is much more memory efficient than AutoHAS. Second, we further extend the co-optimization scope from NAS and training hyper-parameters to also include data augmentation hyper-parameters. To the best of our knowledge, we are the first to jointly optimise all three aspects and we demonstrate the advantage of such practice in Sec. 4.

