# OpenReview forum: "DiffAutoML: Differentiable Joint Optimization for Efficient End-to-End Automated Machine Learning"
_ICLR.cc/2021/Conference — Reject_

### Official Review · AnonReviewer3 · 2020-10-24
**Nice paper for fully differentiable autoML pipeline as long as some concerns on empirical evaluations are resolved.**

**Rating:** 5
**Confidence:** 3

**Review:**

**Summary**
This paper shows how the complex autoML pipeline for neural networks can be trained in an end-to-end manner by combining existing methods. By using backpropagatable discrete sampling methods (Gumbel softmax), input transformed by data augmentation is seamlessly embedded in full backpropagation flow. And a differentiable architecture search algorithm is used, which also incorporates architecture search in full backpropagation flow. On top of this differentiable procedure, an alternating optimization is introduced to train network parameters and hyperparameters.


**Strengths**
1. The paper shows that end-to-end automl is possible in a fully differential way which allows better models with fewer resources.
2. Even though, intuitively, this joint optimization should perform better than non-joint ones, making such training stable does not seem trivial. The authors appear to successfully have taken care of it.


**Weaknesses**
1. Even though 2 hyperparameters are considered in the ablation study, its capability and stability with a moderate number of hyperparameters are not well-addressed (in the main experiment only 1 hyperparameter is considered). And the use of a validation set only in hyperparameter update step make me question that this hyperparameter optimization component may cause some issues such as training instability. Can the authors share some thought, experience, and intuition on this?
2. It is questionable whether the comparison in Figure 2 is fair since DiffAutoML utilizes validation data in its hyperparameter optimization step whereas DSNAS seems to not use validation data.
3. Similar to above, the numbers for DSNAS in Table 1 seem to be taken from DSNAS paper (Table 3) which are numbers from the validation set.


**Recommendation**
The paper tackles interesting and practical problems and shows the proposed method outperforms baselines. My main concern is whether DiffAutoML uses validation data that is not used by other baselines, which could be a reason for better performance. The concern on considering on a small number of hyperparameters is a minor one if some explanation can be added. As long as the concern on the validation data access is resolved then I will be willing to increase my score favoring acceptance. However, in its current form, I cannot stand for it.


**Questions**
- The numbers reported in Table 1 and Table 2 are test performances?


**Additional feedback** (Irrelevant to the decision assessment)
- Since the most of NAS component in the full pipeline is from DSNAS, it would be better for readers if this is detailed in Related Works or Appendix.
- At 4th line in Algorithm 1, using the word 'parent' network is more consistent?
- In the last line in Abstract, en-to-end -> end-to-end
- At 3 lines above eq. 3. ont-hot -> one-hot
- In the last paragraph in Introduction, in the 4th line from the last, Differentiable -> differentiable

---

> ### Author Response · Authors · 2020-11-24
> **Response to AnonReviewer3**
>
> Thanks for your detailed and insightful comments. All code will be
> released after acceptance.
>
> **R3.1 (Limited Hyper-parameters Are Considered and The Validation
> Set)**: (1) Our algorithm can be applied to diverse continuous
> differentiable hyper-parameters including learning rate, momentum,
> weight decay, etc. (2) The main reason why we use training loss for NAS
> is to follow the practice in DSNAS so as to ensure fair comparison. (3)
> In DiffAutoML, we mainly consider the hyper-parameters in the optimizer
> which influence the weights updating process as in [2]. The performance
> of updated weights in the validation set can be a direct evaluation of
> the hyper-parameter setting.
>
> [1] Revisiting the Train Loss: an Efficient Performance Estimator for
> Neural Architecture Search.
>
> [2] Online Learning Rate Adaptation With Hypergradient Descent.
>
> **R3.2 (DSNAS Not Using Validation Set)**: Thanks for pointing it out.
> We would like to clarify that our comparison is still fair as we haven’t
> used an extra validation data. Our validation set is sampled from the
> training set used by DSNAS. For the experiment on ImageNet, we hold out
> $5$% of the ImageNet training data as the validation data.
>
> **R3.3 (Results Are Taken From DSNAS)**: To make a fair comparison, we
> test the performance of DSNAS in the test set. The difference between
> our reported results and the reported results in DSNAS is shown in Table
> 1.
>
> **R3.4 (More Details in Appendix)**: We would include a more detailed
> explanation of related works in our Appendix.
>
> **R3.5 (Parents Network In Algorithm and Other Typos)**: Yes, thanks for
> pointing them out. We will modify our typos in the camera-ready version.

---

### Official Review · AnonReviewer4 · 2020-10-26
**DiffAutoML: Differentiable Joint Optimization for Efficient End-to-End Automated Machine Learning**

**Rating:** 4
**Confidence:** 4

**Review:**

Summary and contributions:
This paper proposes a joint differential search method to optimize data augmentation, discrete architecture choices, and hyperparameters.


Strengths:
1). This paper proposes an automated method to search for data augmentation, hyperparameters, and architectures using gradient descent, which is simple and easy to integrate.
2). It achieves co-optimization on different components in one-stage.
3). A modest improvement upon SOTA mobile models.

Weaknesses:
In the optimization, the proposed method naively uses validation data to optimize the hyperparameters, and use training data to optimize architecture weights, data augmentation weights, and network weights. Wouldn't the architecture choices and data augmentation overfit to the training data in such case? Note that there exists a discrepancy in the optimization direction between NAS and  DAS, where NAS aims to minimize the training loss (e.g. cross entropy) while DAS tends to increase the training loss. Therefore, trying to optimize NAS and DAS under a single objective is intractable. The objectives for the jointly optimization needs to be carefully designed so that the optimization for NAS and DAS can contribute collaboratively to the final testing performance.


Correctness: Are the claims and methods correct? Is the empirical methodology correct?
I am uncertain the method appears to be correct.

Clarity: Is the paper well written?
Yes, although a few aspects could be improved (see feedback).

Relation to prior work: Is it clearly discussed how this work differs from previous contributions?
Yes.

Reproducibility: Are there enough details to reproduce the major results of this work?
No code is provided.

Additional feedback, comments, suggestions for improvement, and questions for the authors:
Clearly describe mathematically how the proposed method contribute collaboratively to the final testing performance including its generalizability. The mutual dependency between NAS and DAS arises from that NAS is based on the training data modified by the augmentation policy generated by DAS and the DAS is based on the network generated by NAS. Additionally, provide ablation studies on changing one single component while fixing others.

---

> ### Author Response · Authors · 2020-11-24
> **Response to AnonReviewer4**
>
> Thanks for your detailed and insightful comments. All code will be
> released after acceptance.
>
> **R4.1 (Different Loss Function)**: The main intuition behind optimizing
> NAS through decreasing the training loss is that, during the early
> training stage, sum over training losses (SoTL) are correlated with the
> final architecture’s performance. Lower SoTL indicates better final
> performance of evaluated neural architecture, thus the optimization of
> NAS is realized through the training loss [1]. It is a common practice
> to use training loss in NAS [2-3], suggesting that this would not result
> in the over-fitting problem. As for data augmentation, using training
> loss has already been explored in previous works [4]. By increasing the
> training loss, we intend to find those transformation strategies which
> are hard for current network. Through paying more attention to
> ’difficult’ transformation, we increase the generalizability of our
> model.
>
> [1] Revisiting the Train Loss: an Efficient Performance Estimator for
> Neural Architecture Search.
>
> [2] SNAS: Stochastic Neural Architecture Search.
>
> [3] DSNAS: Direct Neural Architecture Search without Parameter
> Retraining.
>
> [4] Adversarial Autoaugment.
>
> **R4.2 (Mathematical Description & Ablation Studies)**: Thanks for your
> suggestion. This paper concentrates on proposing a co-optimization
> algorithm for NAS, DA, and HPO. Mathematical explanation about the
> coupling relation between different modules like NAS and DA would be
> considered in our future research. We provide several ablation studies
> in our paper, including realizing the co-optimization in a sequence
> shown in Table 2 upper part, realizing the DA and HPO in the fixed
> network architecture shown in the lower part of Table 2.

---

### Official Review · AnonReviewer1 · 2020-10-27
**Incremental paper with limited results**

**Rating:** 4
**Confidence:** 4

**Review:**

Some of the choices that have to be made when training a neural net based image model are: type of data augmentation, architecture of the neural network, and other hyperparameters such as regularization and optimization hyperparameters (e.g. learning rate). Optimizing all of these is a challenging problem, NAS deals with architecture but ignores the others. More general hyperparameter optimization techniques such as Bayesian Optimization struggle with the dimensionality of the architecture parameters. And optimizing them independently might lead to local minima, and/or be slow.

One previous work (AutoHAS) already combined NAS with the ability to add differentiable hyperparameter. In this work data augmentation is incorporated as well and tuned jointly with the rest of the architecture and the other hyperparameters.

**Their main contributions** of this paper are: 1. The authors describe a NAS+differentialble hyperparameter tuning technique that also include data augmentation, thus extending the setting somewhat compared to previous results 2. They provide experimental results on ImageNet

The paper is reasonably clear, and while some of the ideas are interesting (such as how to incorporate data augmentation as a differentiable NAS hyperparameter) I don't think this work should be accepted mainly because:
* They describe only incremental improvements. Thus no major contribution from the theory/new tecniques part of the paper.
* Limited experimental results, definitely not strong enough to compensate for the previous point. There is no comparison to AutoHAS, which appears to be the closest method, according to the paper itself. And most results are on ImageNet only. If the paper wants to provide a technique for automatically generating vision pipelines, stronger arguments and experimental results should be brought forward for why this will work on new problems and not just on ImageNet (for witch we can just take one of the existing pretrained models).


##### Questions/comments:

In the results, it seems that the competing method use no data augmentation at all, is that correct? If so it would be more fair if the competing method use untuned but reasonable data augmentation.

Does baseline in table 2 contain any data augmentation? It's not really clear?

The results are the average over how many repetitions? It would be nice to know the number of repetitions and standard dev/error, or if a single one at least have the source code available for better reproducibility.

The validation set is used for selecting hyperparameters, thus should not be used for comparing methods, was a separate test set used for accuracy? It would be good to describe this more clearly in the paper.

I would remove the epoch based plot, what we care about at the end is time.


##### Typos:

Section 1
"We first jointly optimize three different AutoML components, including" -> you might want to remove "including" if you list all three
Section 3.1
For t-th iteration -> for the t-th iteration

---

> ### Author Response · Authors · 2020-11-24
> **Response to AnonReviewer1**
>
> Thanks for your detailed and insightful comments. All code will be
> released after acceptance.
>
> **R1.1: Incremental Results**: Compared with previous methods,
> DiffAutoML is the first work showing that \*\*end-to-end\*\* AutoML is
> possible in a fully differential way. This problem setting is practical
> but rarely explored before. Specifically, DiffAutoML achieves joint
> optimization of different AutoML components, including neural
> architecture search, data augmentation, and hyper-parameter
> optimization, and finds better ready-to-use models with fewer computing
> resources. In the experiments, we show around one percent increase on
> ImageNet compared with existing end-to-end AutoML algorithms.
>
> **R1.2: Other Computer Vision Tasks**: The main reason why we have not
> compared DiffAutoML with AutoHAS is that AutoHAS requires much more
> computational resources than DiffAutoML and AutoHAS cannot directly
> realize the hyper-parameter optimization of continuous hyper-parameters
> which is discretized during the optimization process in AutoHAS. More
> specifically, at each iteration, AutoHAS would discretize continuous
> hyper-parameter into a linear combination of discrete hyper-parameters.
> And those discrete hyper-parameters will be applied to optimize the
> network. During this process, AutoHAS has to deep copy the network for N
> times and N is the number of discretized hyper-parameters. Thus,
> directly comparing AutoHAS with DiffAutoML is unfair. ImageNet is the
> most popular benchmark used to evaluate machine learning algorithms,
> which is representative and credible. In order to prove the
> generalizability of DiffAutoML, we have applied our model on CIFAR-10
> and the results are shown in **R2.1**.
>
> **R1.3 (Data Augmentation In Other NAS)**: Existing NAS algorithms
> listed in Table 1 require additional human intervention and
> computational cost, such as deciding on the hyperparameters for model
> re-training or choosing the data augmentation policy. On the other hand,
> DiffAutoML provides an end-to-end solution to achieve NAS and automated
> DA simultaneously with few additional human involvement. In order to
> make fair comparison, we have adapted the random augmentation with
> DiffAutoML, where the data augmentation strategies are sampled from
> uniform distribution same as their level instead of being learned from
> the training loss function. The results are shown in the following
> Table S2. Moreover, in Table 2, we applied automated DA and HPO to previous
> NAS algorithms, including Darts and DSNAS (see row 2-3, column 3 in
> Table 2). As can be seen, DiffAutoML still outperforms these baselines.
>
> Table S2: Performance of Random-DiffAutoML and DiffAutoML on ImageNet Dataset.
>
> | Algorithm      | Flops | Search Stage(%)    |
> | :---        |    :----:   |          ---: |
> | Random-DiffAutoML      | 321M       | 74.8/92.1   |
> | DiffAutoML  | 318M        | 75.5/92.7      |
>
> **R1.4 (Repetition of Experiments)**: All our experiments have been run
> for more than twice. As we have not noticed evident variance between
> different experiments, we reported the mean test accuracy in Table 1.
> For the cases with evident variance, we also reported the standard
> deviation, together with the mean test accuracy, in Table 2. All the
> source code and well-trained model will be released, once our paper has
> been accepted.
>
> **R1.5 (Validation Set)**: Thanks for pointing it out. To make a fair
> comparison, we haven’t used an extra validation set. Our validation set
> is sampled from the training set. For the supplementary experiment on
> CIFAR-10, we have adopted the training set and validation set split
> given by ’Darts: differentiable architecture search’, where they hold
> out half of the CIFAR-10 training data as the validation data. For the
> experiments on ImageNet, we hold out $5$% of the ImageNet training data
> as the validation data.
>
> **R1.6 (Figure 2)**: In Figure 2, we would like to show that even though
> our model needs more time for one epoch compared with pure NAS, due to
> additional HPO and automated DA, our model can converge within fewer
> epochs and thus saves the total computation cost.
>
> **R1.7 (Typos)**: Thanks for pointing it out, we will modify the typos
> for the revised manuscript.

---

> > ### Comment · AnonReviewer1 · 2020-11-24
> > **Validation set**
> >
> > Hi, thanks for you answer. For now I just to clarify one detail that is still not fully clear to me in your answer. What validation set exactly was used for the results in the tables? And what is used to compute the validation loss in DiffAutoML? Are these the different sets or are they the same?

---

### Official Review · AnonReviewer2 · 2020-10-29
**Overall this paper is clearly written and easy to follow. The motivation makes sense and result support the claims.**

**Rating:** 6
**Confidence:** 3

**Review:**

Summary
This paper focuses on achieving automated "from data to model" including different components in modeling, namely data augmentation, Neural Architecture Search, Hyper Parameter Optimization. The proposed approach first use data augmentation to select the data argumentation transformation. It tries to select examples which incurs higher training loss for the model to address hard examples. Then use the DAG for neural architecture search. Given the data and architecture, it then alternatively update the model parameter and hyper parameter. The overall proposed framework is end-to-end. Experiment on ImageNet shows slight performance improvement over existing approaches. The authors also conduct ablation study to show the effectiveness of jointly modeling the three components (data augmentation, neural architecture search, hyper-parameter optimization).

Strengths:
1) This work considers three important components in modeling process including data argumentation, HPO, and NAS. The different components may interact with each other to impact the performance.
2) This paper is clearly written and easy to follow.
3) The experiment shows performance improvement with less time needed.

Weakness
1) The experiment is limited with only 1 task and data. Is it possible to show the results on more than 1 dataset? Otherwise, the result might still be dataset specific. However, the proposed approach should not be

Questions
Will the proposed approach generalize beyond computer vision task?
Why are the data augmentation and neural architecture search grouped together? I was wondering what will happen if you group neural architecture search and hyper parameter optimization first?
How should I interpret figure 3? How could you demonstrate that the selected architecture by the proposed approach is better or make sense?

---

> ### Author Response · Authors · 2020-11-24
> **Response to AnonReviewer2**
>
> We appreciate your constructive suggestion and we provide the following
> modification. All code will be released after acceptance.
>
> **R2.1 (More Datasets)**: Our DiffAutoML can be easily applied to other
> search spaces based on weight sharing strategy, e.g., the search space
> proposed in ’Path-level network transformation for efficient
> architecture search.’. To illustrate the generalizability of DiffAutoML,
> we have applied DiffAutoML to CIFAR-10 with the search space in ’Darts:
> differentiable architecture search’. In the first experiment, we applied
> the co-optimization in a sequence (Pipeline-DiffAutoML). We first apply
> DARTS to find the structure and then, based on this structure the data
> augmentation strategy and the hyper-parameter optimization strategy are
> applied to tune this learned structure. In the second experiment, we
> directly applied DiffAutoML to CIFAR-10 (DiffAutoML). The average
> results over x seeds are shown in Table S1. With a different search
> space, DiffAutoML still performs better than optimising different
> modules in sequence (Pipeline-DiffAutoML).
>
> Table S1: Performance of DiffAutoML on CIFAR-10 dataset.
>
> | Algorithm      | Params | Test Error(%)     |
> | :---        |    :----:   |          ---: |
> | Darts      | 3.3M       | 2.80   |
> | Pipeline-DiffAutoML  | 3.3M        | 2.47      |
> | DiffAutoML  | 3.3M        | 2.39      |
>
> **R2.2 (Other Computer Vision Tasks)**: DiffAutoML is a general joint
> optimization framework that can be applied to different tasks, such as
> object detection, semantic segmentation, or even NLP tasks. The designs
> of the losses for automated DA, NAS, and HPO are task-agnostic. For a
> new task, we only need to determine the suitable search spaces for DA
> and NAS, and then our method can be applied to this task directly.
> ’Auto-DeepLab: Hierarchical Neural Architecture Search for Semantic
> Image Segmentation’ proposes a two-level search space to deal with the
> image segmentation task and the gradient-based method, e.g., Darts and
> DiffAutoML, can be used to realize this kind of neural architecture
> search.
>
> **R2.3 (Grouping NAS and DA)**: Thanks for your question. We haven’t
> grouped neural architecture search and data augmentation deliberately.
> The common thing in the neural architecture search and data augmentation
> is that both of them use the training loss for optimization. The
> optimization for neural architecture search aims to decrease the
> training loss for the sake of finding better structures, while the
> optimization for data augmentation aims to increase the training loss to
> let the training process pay more attention to the hard samples.
> Moreover, during the optimization process, different optimizers are used
> for updating neural architecture parameters and data augmentation
> parameters. Thus there two processes wouldn’t conflict with each other.
> Different AutoML components, including NAS, HPO, and Automated DA, are
> optimized jointly in a unified framework.
>
> **R2.4 (Figure 3)**: With Figure 3, we would like to show that with
> different training settings, we can end up with different optimal neural
> architectures. Thus, it is necessary to consider the co-optimization of
> different AutoML components together, instead of performing them
> one-by-one sequentially.
>
> **R2.5 (How To Compare)**: The superiority of the found architectures is
> mainly demonstrated by comparing their final accuracy, their
> computational request, and their model size on ImageNet with those of
> other competitors.

---

### Decision · Program_Chairs · 2021-01-07
**Final Decision**

**Decision:**

Reject

**Comment:**

The paper was discussed by the reviewers that acknowledged the rebuttal and the authors’ responses. In particular, they appreciated the fact that some of their concerns were alleviated (e.g., going beyond the single ImageNet evaluation).

More generally, while all the reviewers thought that the problem tackled by the paper was of clear interest (i.e., full end-to-end auto-ML encompassing DA, NAS and HPO), they still expressed concerns (even after the rebuttal), in particular:

* _Clarity of the methodology_: None of the reviewers could clearly and fully understand the mathematical formulation of the joint optimization, leading to a series of questions regarding the confusing usage of the training/validation set in the experimental setup. This unfortunately made the assessment of (some aspects of) the paper speculative for the reviewers.
* _Comparison with AutoHAS_: AutoHAS and DiffAutoML are obviously related methods. Even if AutoHAS has weaknesses compared to the proposed approach DiffAutoML, e.g., discretization of the continuous hyperparameters and no tuning of DA, it is still meaningful to carry out an actual comparison (possibly normalized by the different costs at play since the authors have highlighted the different memory overheads). Though the listed weaknesses of AutoHAS _should_ play in favor of DiffAutoML,  a proper experimental comparison would better support that claim.

Given those remaining concerns and the overall mixed scores, the paper is recommended for rejection. The detailed comments of the reviewers provide an actionable list of items to improve the paper for a future resubmission.